# Serum Collected from Preeclamptic Pregnancies Drives Vasoconstriction of Human Omental Arteries—A Novel Ex Vivo Model of Preeclampsia for Therapeutic Development

**DOI:** 10.3390/ijms231810852

**Published:** 2022-09-16

**Authors:** Bianca R. Fato, Natasha de Alwis, Sally Beard, Natalie K. Binder, Natasha Pritchard, Stephen Tong, Tu’uhevaha J. Kaitu’u-Lino, Natalie J. Hannan

**Affiliations:** 1Therapeutics Discovery and Vascular Function in Pregnancy Group, Mercy Hospital for Women, University of Melbourne, Heidelberg, VIC 3084, Australia; 2Department of Obstetrics and Gynaecology, Mercy Hospital for Women, University of Melbourne, Heidelberg, VIC 3084, Australia; 3Diagnostic Discovery and Reverse Translation in Pregnancy Group, Mercy Hospital for Women, University of Melbourne, Heidelberg, VIC 3084, Australia

**Keywords:** preeclampsia, esomeprazole, therapeutics, sFlt-1, sENG, ET-1

## Abstract

New-onset maternal hypertension is a hallmark of preeclampsia, driven by widespread endothelial dysfunction and systemic vasoconstriction. Here, we set out to create a new ex vivo model using preeclamptic serum to cause injury to the endothelium, mimicking vascular dysfunction in preeclampsia and offering the potential to evaluate candidate therapeutic interventions. Human omental arteries were collected at caesarean section from normotensive pregnant patients at term (*n* = 9). Serum was collected from pregnancies complicated by preterm preeclampsia (birth < 34 weeks’ gestation, *n* = 16), term preeclampsia (birth > 37 weeks’ gestation, *n* = 5), and healthy gestation-matched controls (preterm *n* = 16, term *n* = 12). Using wire myography, we performed ex vivo whole vessel assessment where human omental arteries were treated with increasing doses of each serum treatment (2–20%) and vasoreactivity was assessed. All pregnant serum treatments successfully drove vasoconstriction; no significant difference was observed in the degree of vasoconstriction when exposed to preeclamptic or control serum. We further demonstrated the ability of esomeprazole (a candidate therapeutic for preeclampsia; 0.1–100 µM) to drive vasorelaxation of pre-constricted vessels (only with serum from preeclamptic patients). In summary, we describe a novel human physiological model of preeclamptic vascular constriction. We demonstrate its exciting potential to screen drugs for their therapeutic potential as treatment for vasoconstriction induced by preeclampsia.

## 1. Introduction

Preeclampsia is a pregnancy complication that affects 3–5% of all pregnancies [1]. It is a leading cause of maternal and neonatal death, resulting in >70,000 maternal deaths and >500,000 fetal and neonatal deaths worldwide each year [2,3]. Preeclampsia is defined by new-onset hypertension after 20 weeks’ gestation, accompanied by one or more of either proteinuria (excess protein in the urine due to kidney injury), placental insufficiency, and/or another major organ dysfunction [4]. The increase in blood pressure is thought to be caused by a decrease in systemic arterial compliance and an increase in peripheral vasoconstriction [5,6,7].

A further classification that infers the severity of the disease has been established; early-onset (preterm) preeclampsia is when delivery is necessitated prior to 34 weeks’ gestation [8]. There is an intersection in the culmination of preterm and term preeclampsia; both result in hypertension and often end-organ injury and/or fetal growth restriction. However, differences in aetiology, risk factors, and outcomes are apparent when they are stratified [9]. Compared to term preeclampsia, there is a higher probability of adverse birth outcomes when preeclampsia arises in preterm gestation.

Inadequate placentation is thought to be associated with severe early-onset preeclampsia [10]; poor remodelling of the spiral arteries leads to perturbations in blood flow to the placenta, resulting in placental dysfunction. The injured and stressed placenta releases antiangiogenic and proinflammatory mediators into the maternal circulation, where they neutralise proangiogenic factors and cause serious widespread endothelial dysfunction and systemic vasoconstrictor effects. Endothelial dysfunction and systemic vasoconstriction can compromise blood flow to major organ systems and the uteroplacental interface [11].

Candidate therapeutics to prevent or treat preeclampsia should aim to mitigate systemic vasoconstriction and endothelial dysfunction experienced by the maternal vasculature. Myography is an ex vivo technique that allows us to test this. Myograph experiments permit assessment of the functional responses and vascular reactivity of small resistance arteries (vessels that contribute to changes in blood pressure). This technique is useful in establishing whether a candidate therapeutic may have a direct effect on the vasculature and could reduce the systemic vasoconstriction associated with the pathophysiology that underlies preeclampsia. Traditionally, myograph studies use chemical or synthetic peptides to illicit vasoconstriction in order to test a candidate drug’s ability to prevent or rescue this response. Whilst these compounds are effective in driving constriction of whole vessels, this approach is limited to inducing constriction through the specific pathway activated by the chemical/synthetic vasoconstrictor and thus, is unlikely to encapsulate the complexity of changes occurring in the vasculature in preeclampsia. In pregnancies complicated by preeclampsia, the maternal blood contains a plethora of factors released by the dysfunctional placenta that contribute to vascular dysfunction, such as antiangiogenic factors (including soluble fms-like tyrosine kinase-1 (sFlt-1) and soluble endoglin (sENG)), vasoconstrictors (including endothelin-1 (ET-1)), and proinflammatory factors (including tumor necrosis factor). These factors are found in high concentrations in the serum of preeclamptic patients [12,13,14,15]. Thus, inducing vasoconstriction of arteries using preeclamptic serum, containing such pathological mediators, may more closely mimic the disease pathology and provide a useful model for drug testing. 

Our team has a long-standing interest in repurposing medications with good safety profiles for the prevention/treatment of preeclampsia. This includes the proton pump inhibitors (PPIs) [16], particularly esomeprazole. We have previously demonstrated that esomeprazole reduced maternal blood pressure in a mouse model of preeclampsia, rescued endothelial dysfunction, and dilated blood vessels in primary human tissues and mouse models [16]. However, the precise mechanisms behind these actions are not yet understood. 

The current study aimed to develop a new vascular ex vivo model using serum collected from preeclamptic patients to determine whether this drives constriction of arteries collected from normotensive pregnancies; to better mimic the preeclamptic phenotype. Further, we aimed to use this model to test the effects of esomeprazole on vascular reactivity in the presence of physiologically relevant preeclamptic mediators. 

## 2. Results

### 2.1. Antiangiogenic Factors sFlt-1 and sENG, and the Vasoconstrictor ET-1, Are Increased in Serum from Pregnancies Complicated by Preeclampsia

Initially, we measured the levels of antiangiogenic factors sFlt-1 and sENG, and the potent vasoconstrictor ET-1, in serum samples via ELISA to select a cohort for myograph experiments. As expected, circulating sFlt-1 was significantly higher in preterm preeclamptic serum (Figure 1A, *p* < 0.0001; preeclamptic *n* = 10, control *n* = 10) and term preeclamptic serum (Figure 1B, *p* = 0.0095; preeclamptic *n* = 4, control *n* = 6) compared to their respective gestation-matched controls. Further, sENG was also significantly increased in preterm preeclamptic serum (Figure 1C, *p* < 0.0001; preeclamptic *n* = 10, control *n* = 10) compared to gestation-matched controls; sENG was not significantly different in pregnancies complicated by term preeclampsia, in contrast to gestation-matched controls (Figure 1D, *p* = 0.1355, preeclamptic *n* = 4, control *n* = 6). Compared to control serum (*n* = 9), the concentration of ET-1 was significantly higher in preterm (*n* = 10) preeclamptic (Figure 1E, *p* < 0.0001) serum. The levels of ET-1 in term preeclamptic (Figure 1F, *n* = 3) serum were not significantly different to gestation-matched control serum (*n* = 6). 

**Figure 1 ijms-23-10852-f001:**
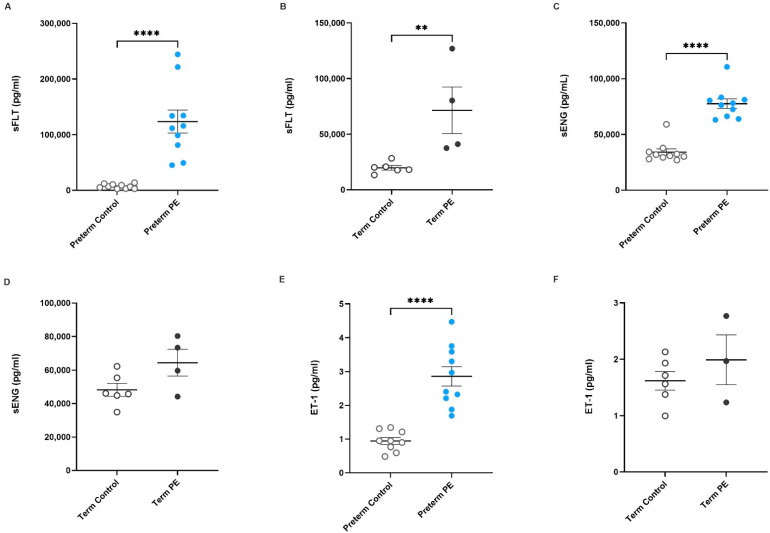
Antiangiogenic factors sFlt-1 and sENG, and potent vasoconstrictor ET-1, are increased in serum obtained from pregnancies complicated by preeclampsia. Circulating sFlt-1 concentrations in serum from pregnancies complicated by (**A**) preterm preeclampsia (
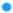
) or (**B**) term preeclampsia (
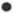
) and from patients with uncomplicated pregnancies at matched gestation (preterm and term controls; 
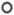
) are shown. Circulating sENG concentration in (**C**) preterm and (**D**) term preeclamptic serum in comparison to gestation-matched controls are also presented, as well as ET-1 levels in serum collected from (**E**) preterm and (**F**) term preeclamptic pregnancies and gestation-matched controls. Individual points represent individual patients, and the data are expressed as mean ± SEM (*n* = 4–10). *p*-values indicated as follows: ** *p* < 0.01, **** *p* < 0.0001.

### 2.2. Serum from Pregnancies Complicated by Preeclampsia Did Not Induce Greater Constriction of Human Omental Arteries Compared to Serum from Gestation-Matched Controls

We next assessed whether maternal serum collected from normotensive pregnancies and pregnancies complicated by preeclampsia could directly induce changes in vascular reactivity in omental arteries. We observed that both control (preterm and term) and preeclamptic serum successfully induced vasoconstriction of the omental arteries. Notably, there was no significant difference in the percentage of vasoconstriction between the preeclamptic and control treatments at any concentration of serum for both the preterm (Figure 2A; preeclamptic *n* = 6, control *n* = 5) and term (Figure 2B; preeclamptic *n* = 4, control *n* = 4) groups. In these assessments, we investigated the area under the curve that defines the total constriction across the concentrations of serum used; the area under the curve was not different between preeclamptic and control serum (Figure 2C,D). The EC50 value shown in Figure 2E,F describes the concentration of serum treatment required to induce 50% of the maximum constriction. There was no difference in the EC50 value produced by either control or preeclamptic serum administration. The maximum constriction plots (Figure 2G,H) further confirmed that there was no difference between the constrictions induced by preeclamptic or control serum. These data demonstrate that serum collected from pregnancies can be used to induce vasoconstriction of omental arteries, and that this can enhance ex vivo assessment of candidate therapeutics.

### 2.3. Esomeprazole Treatment Induced Vasodilation of Pregnant Human Omental Arteries Pre-Constricted with Serum Collected from Preterm Preeclamptic Patients

Given our interest in the PPI esomeprazole as a potential candidate treatment for preeclampsia, we next assessed its potential in our new model of vascular dysfunction. Specifically, we assessed the ability of esomeprazole to mitigate the vasoconstriction induced by serum from both preterm preeclamptic pregnancies and normotensive gestation-matched control pregnancies. We observed that, in arteries constricted with preterm preeclamptic serum, there was no significant difference in the percentage of relaxation with increased concentrations of esomeprazole when compared to the vehicle control (Figure 3A, *n* = 3). However, analysis of the area under the curve demonstrated a significant difference between esomeprazole and vehicle-treated vessels (Figure 3B; *p* = 0.039; *n* = 3). The increased area under the curve presented in Figure 3B indicates there was a significant increase in total vasorelaxation with esomeprazole treatment. Further, when maximum relaxation data were analysed, esomeprazole treatment resulted in a significant increase in relaxation in arteries constricted with preterm preeclamptic serum (Figure 3C, *p* = 0.0304), compared to those treated with vehicle control. 

Following this, we next examined whether there were differences in vasoreactivity in omental arteries constricted with gestation-matched, preterm control serum. Of note, in contrast to the experiments using serum from pregnancies complicated with preeclampsia, there was no difference in the percentage of relaxation (Figure 3D), the area under the curve (Figure 3E), or maximum relaxation (Figure 3F) in arteries treated with esomeprazole when constricted with preterm control serum.

**Figure 2 ijms-23-10852-f002:**
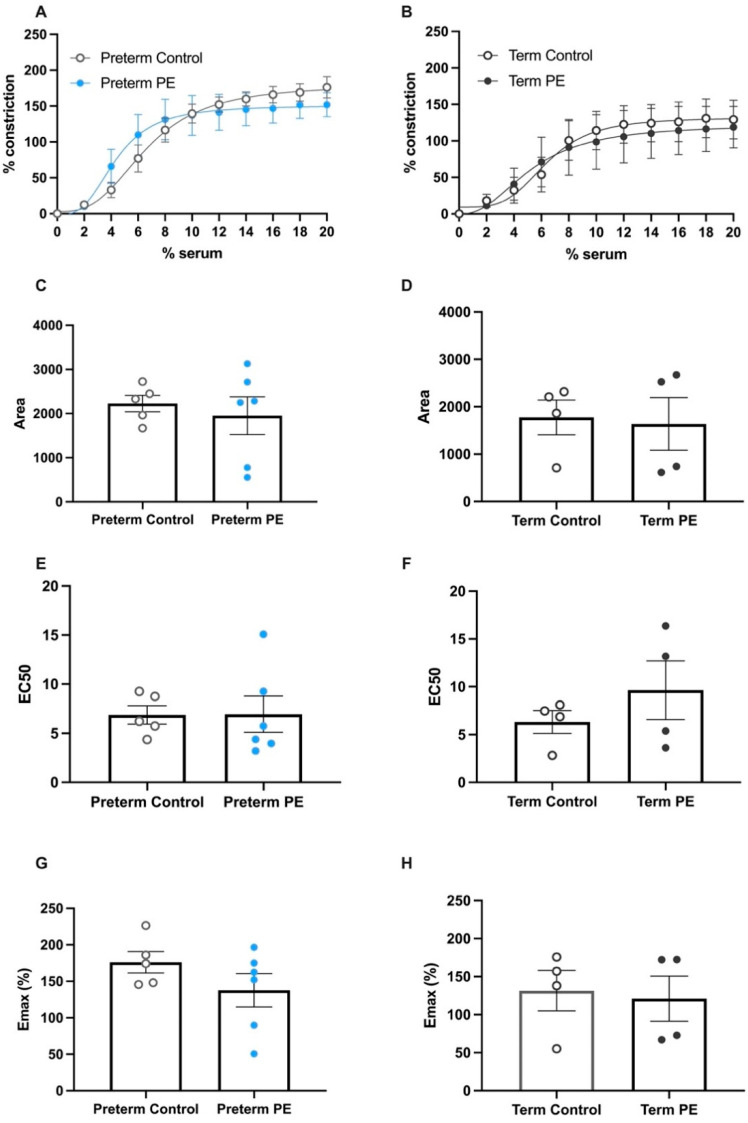
Serum from patients with preeclampsia does not induce greater vascular constriction of human omental arteries compared to serum from gestation-matched controls. Constriction curves of healthy (normotensive) term human omental arteries after being constricted by serum from (**A**) preterm preeclampsia (
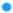
) and (**B**) term preeclampsia (
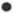
) are compared to gestation-matched preterm and term controls (
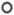
), respectively. The percentage of constriction was normalised to the maximum constriction induced by 50mM potassium physiological salt solution (KPSS). Analysis of the area under the curve in (**C**) preterm and (**D**) term constriction curves is shown; (**E**) preterm EC50 and (**F**) term EC50 values are derived; and maximum constriction (E_max_) in response to (**G**) preterm serum and (**H**) term serum compared to gestation-matched controls is presented. Individual points represent individual patients (*n* = 4–6). The data are expressed as mean ± SEM.

**Figure 3 ijms-23-10852-f003:**
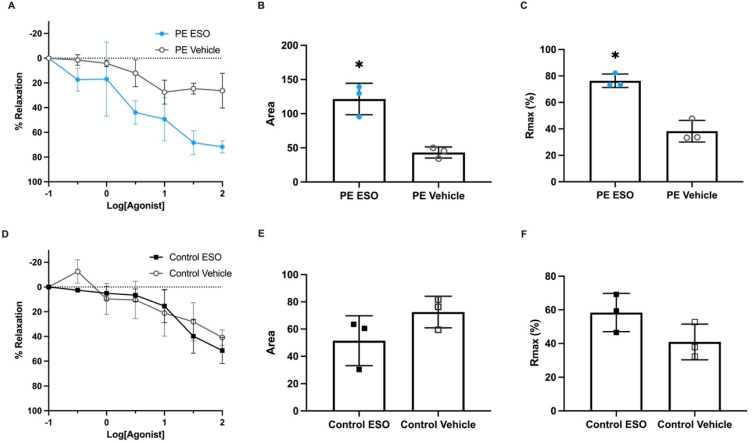
Esomeprazole treatment relaxed human omental arteries constricted with serum collected from preterm preeclamptic pregnancies, compared to the vehicle (control). Vascular relaxation curves of human term omental arteries (normotensive) constricted by (**A**) serum obtained from pregnancies complicated by preterm preeclampsia, with either esomeprazole or vehicle treatment (
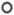
), are presented. The percentage of relaxation was normalised to the maximum relaxation induced by bradykinin. Analysis of the area under the curve (AUC) (**B**) and maximum relaxation (**C**) in response to esomeprazole (
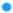
) or the vehicle (
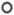
) showed arteries treated with esomeprazole had enhanced relaxation when constricted with serum from preterm preeclampsia. Omental arteries constricted with preterm control (gestation-matched) serum: relaxation curves (**D**), as well as AUC (**E**) and maximum relaxation plots (**F**), demonstrate there was no significant effect with esomeprazole treatment (
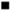
) compared to vehicle control (□). Individual points represent individual patients (*n* = 3). Data are expressed as mean ± SEM; *p*-values indicated as follows: * *p* < 0.05.

## 3. Discussion

Damaged and dysfunctional maternal vasculature underpins the significant obstetric condition of preeclampsia. Important preclinical molecular studies by our group and others, have identified markers of oxidative stress and upregulation of inflammatory and antiangiogenic factors by the preeclamptic placenta [17,18,19]. Many of these factors are also found in excess in the maternal circulation in pregnancies complicated by preeclampsia, where it has been suggested that they contribute to serious endothelial and vascular dysfunction. This study aimed to investigate the effect of maternal serum, a constituent of maternal whole blood, on resistance arteries obtained from pregnant patients. Here, we describe a new ex vivo whole vessel model of preeclampsia, which can be used to determine the effectiveness of novel therapeutics in relation to combatting the vascular pathophysiology underlying preeclampsia. Overall, we identified that pregnant serum induced vasoconstriction of maternal omental arteries. Further, the proton pump inhibitor esomeprazole dilated arteries constricted with serum obtained from pregnancies complicated by preterm preeclampsia.

The predominant objective of this paper was to determine whether serum could induce vasoconstriction of normotensive pregnant arteries. Most studies assess vascular changes induced by plasma; in these assessments, plasma has been used to incubate vessels prior to myograph assessment, where the arteries’ vasoreactivity to constriction and dilation compounds was assessed. Hence, the effect of preeclamptic serum on resistance arteries is not well understood. Reportedly, plasma collected from pregnancies complicated by preeclampsia induced a significant reduction in endothelium-dependent relaxation in myometrial vessels from normotensive pregnant patients [20,21], but did not alter endothelium-independent vascular smooth muscle function [22]. Serum has a similar composition to plasma in that it contains proteins, immunoglobulins, antigens, and exogenous substances; it is a source of a plethora of soluble mediators commonly utilised in diagnostic tests as biomarkers [23]. In contrast to plasma, however, serum is devoid of coagulants and factors which promote clotting [24,25]. Hence, for this model, we can avoid the addition of anticlotting factors such as heparin, which can alter vascular reactivity itself in the myograph chamber.

Here, we evaluated serum from pregnancies complicated by preeclampsia or gestation-matched, normotensive control pregnancies in relation to inducing vasoconstriction. It is well established that sFlt-1 and sENG are significantly elevated in the circulation of pregnancies complicated by preeclampsia [12,26,27,28,29], and their levels correlate with disease severity [28]. ET-1 has also been shown by our team and others to be significantly elevated in circulation preceding diagnosis of preeclampsia [14,30,31] and following diagnosis. Excess levels of sFlt-1 and sENG in the circulation is thought to inflict injury to the endothelial cells lining the maternal vasculature, which is an important consideration when considering the pathophysiology that occurs, and should be a focus when developing approaches to treating preeclampsia [32,33]. Here, as expected, we confirmed that serum obtained from pregnancies complicated by preterm preeclampsia had significantly higher concentrations of sFlt-1, sENG, and ET-1 compared to gestation-matched controls. Only sFlt-1 was significantly increased in the term preeclamptic samples assessed, with trends of an increase observed for sENG and ET-1. Increased sample size would have also likely yielded a significant increase in term preeclampsia [14,34].

Contrary to our expectations, when we assessed vasoreactivity in response to serum obtained from either the preeclamptic or control cohorts, we observed that both induced similar vasoconstriction of human omental arteries. These data demonstrate that serum collected from pregnant patients, both term and preterm, could be used to induce vasoconstriction in ex vivo myography experiments. Whilst both serum treatments induce similar vasoconstriction in this model, we propose that serum obtained from pregnancies complicated by preeclampsia is more physiologically relevant, given the aberrant levels of antiangiogenic, vasoconstrictive, and proinflammatory molecules [12,13,14,15,28,35]. These mediators contribute to serious endothelial and vascular dysfunction in preeclampsia, driving further injury to the vasculature.

This study utilised human omental arteries, resistance arteries which contribute to changes in blood pressure, and are used to model systemic vascular changes. We believe that this is the best vessel type we have available to address the systemic vasoconstriction and hypertension featured in preeclampsia. Further, omental arteries are relatively easy to access, being collected during caesarean section. However, assessing the effect of serum collected from pregnancies on other vessels, including myometrial and chorionic arteries, may also be of interest when identifying local organ effects on the uterus and placenta. Assessing the effect of serum on mouse/rat systemic mesenteric arteries may give insight into the differences between human and rodent vasculature—of relevance to animal models of preeclampsia.

Esomeprazole has previously been shown to dilate blood vessels in primary human tissues and other mouse models of preeclampsia. Its actions can be attributed to its ability to reduce secretions of antiangiogenic factors preclinically, including sFlt-1 and sENG [16,36], and to ameliorate endothelial dysfunction, which could indirectly enhance endothelial-dependent relaxation pathways. Importantly, here we demonstrate for the first time that omental arteries constricted using serum from pregnancies complicated by preeclampsia have a different response profile to vessels constricted using serum from normotensive controls when treated with the proton pump inhibitor esomeprazole; arteries constricted with serum from pregnancies complicated by preeclampsia demonstrated a greater relaxation profile. Thus, we suggest using serum from preeclampsia is more physiologically relevant. This further confirms that esomeprazole provides vasoprotective activity in our vascular preclinical models of preeclampsia. Given its safety profile, esomeprazole offers important potential as a therapeutic candidate. Whilst not assessed in the current study, we hypothesise that term serum would act similarly; however, further investigations into differences in constriction need to be made. 

Whilst these novel preclinical models of preeclampsia provide physiologically relevant assays for the assessment of therapeutic effects, we recognise there may be difficulties for other researchers looking to establish this model in their own laboratories, especially given that collection of serum from early-onset preeclamptic patients is only attainable within specific clinical settings. Additionally, a significant amount of serum is required to have sufficient power in the study. Further investigations into the molecular pathways that drive the constrictions induced by serum from pregnancies complicated by preeclampsia, in contrast to gestation-matched controls, would provide further useful insight. This could better identify strong targets for novel prevention therapies against vascular dysfunction and constriction in preeclampsia. Whilst we suggest the use of serum collected from preeclamptic cases is the most physiologically relevant assay, we acknowledge that control serum may remain relevant and useful in other contexts for therapeutic analysis; pregnant serum may be better than chemical/synthetic vasoconstrictor compounds. 

An additional consideration is that the arteries used were obtained from term normotensive pregnancies. Thus, it is important to consider whether these vessels have the same vasoactive potential compared to vessels from pregnancies complicated by preeclampsia or hypertension. Assessment of the distribution and abundance of various receptors in these arteries would add further insight into the complex vasoactivity occurring. 

Finally, wire myograph assays were used to determine vascular reactivity, whereby treatment is added into the chamber and the whole vessel is immersed. Thus, the vascular smooth muscle layer, as well as the inner endothelial lining, is bathed in the treatment/s. In the physiological state, serum is in direct contact with the endothelium and not the outer vascular smooth muscle cell layer. To address whether this may have distinct impacts, another myograph technique known as pressure myography, may have a useful application. Specifically, treatment can be added via cannulation to flow through the lumen of the artery and, thus, not directly act on the vascular smooth muscle layer.

Pregnant serum acts as a physiological stimulant to drive vasoconstriction in vessels obtained from pregnancy. Although serum from both pregnancies complicated by preeclampsia and normotensive control pregnancies drove vasoconstriction of arteries, we importantly demonstrate the physiological difference in the vascular response to the small molecule drug esomeprazole, providing important support for the utilisation of pathophysiological models. Finally, esomeprazole was able to reduce vascular constriction, driving arteriole relaxation in this new ex vivo model of preeclampsia.

## 4. Materials and Methods

### 4.1. Tissue Collection

This study was approved by the Mercy Human Research Ethics Committee (HREC/R11-34, and HREC/R14-11; Mercy Health, Victoria, Australia). For the collection of maternal serum and omental fat samples during pregnancy and following caesarean section delivery, informed written consent was obtained from participating patients presenting to the Mercy Hospital for Women (Heidelberg, Victoria, Australia) prior to sample collection.

#### 4.1.1. Serum Collection

Maternal whole blood samples were collected from patients whose pregnancies were complicated by early-onset preterm preeclampsia (delivery < 34 weeks’ gestation; *n* = 16) or late-onset preeclampsia (delivery > 37 weeks gestation; *n* = 5), as well as from patients with uncomplicated pregnancies at matched gestation (preterm controls *n* = 16; term controls *n* = 12). To provide clear distinction between early- and late-onset preeclampsia, pregnancies delivered between 34 and 37 weeks’ gestation were excluded from this study. Preeclampsia was defined according to the International Society for the Study of Hypertension in Pregnancy (ISSHP) guidelines published in 2018 [4]. Blood (8 mL) was collected into serum tubes (Serum Separator Tube (SST^®^); gold cap), gently inverted, and centrifuged at 1500 g for 10 min. The serum fraction was collected and stored at −80 °C until further assessment. Table 1 summarises patient characteristics for the serum samples collected.

#### 4.1.2. Omental Fat Tissue Collection

Omental fat biopsies were stored in Ca^2+^-free Krebs physiological salt solution (NaCl 120 mM, KCl 5 mM, MgSO_4_ 1.2 mM, KH_2_PO_4_ 1.2 mM, NaHCO_3_ 25 mM, D-glucose 11.1 mM) overnight (16 ± 2 h) at 4 °C to wash out anaesthetics from surgery. Omental arteries were dissected from these samples and used in wire myograph experiments as described below. 

### 4.2. Enzyme-Linked Immunosorbent Assay (ELISA)

The concentration of circulating antiangiogenic factors sFlt-1 and sENG, and the potent vasoconstrictor ET-1, were measured in maternal serum collected from patients diagnosed with early-onset preterm preeclampsia (*n* = 10), term preeclampsia (*n* = 4), and normotensive gestation-matched control patients (*n* = 17). Serum levels were measured using the Human VEGFR1/Flt-1 DuoSet ELISA (1/20 dilution; DY321B, R&D systems, Minneapolis, MN, USA), the Human Endoglin/CD105 DuoSet ELISA (1/15 dilution; DY1097, R&D systems), and the Endothelin-1 Quantikine ELISA Kit (neat serum; DET100, R&D systems, sensitivity: 0.207 pg/mL), according to the manufacturer’s instructions (intra-assay coefficients of variations under 10%). 

### 4.3. Assessment of Vascular Reactivity

Human omental arteries 2 mm in length were carefully dissected from surrounding connective and adipose tissue in Krebs physiological salt solution. The dissected arteries were then mounted on the 620 M Wire Myograph (Danish Myo Technology, Hinnerup, Denmark) chambers with 40 µm tungsten wires (Danish Myo Technology). Chambers were filled with Krebs salt solution, heated to 37 °C, and bubbled continuously with carbogen (95% oxygen, 5% carbon dioxide) to buffer the solution. Arteries were normalised to 100 mmHg (13.3 kPa, IC/IC100 = 1) pressure using the DMT normalisation module on LabChart software (ADInstruments, Sydney, NSW, Australia).

Assessment of the vascular reactivity of each vessel was performed as follows. Vessel smooth muscle reactivity was assessed using high potassium physiological salt solution (KPSS; NaCl 25 mM, KCl 50 mM, MgSO_4_ 1.2 mM, KH_2_PO_4_ 1.0 mM, NaHCO_3_ 25 mM, D-glucose 11.1 mM, CaCl_2_ 2.5 mM). Endothelial integrity was assessed by pre-constricting vessels to the calculated 50–70% of KPSS constriction with a known vasoconstrictor of omental arteries (a thromboxane agonist), U46619 (Sapphire Bioscience, Redfern, NSW, Australia). This was followed by adding a bolus of the endothelial-dependent vasodilator bradykinin (Sapphire Bioscience). Vasorelaxation of 80% was required for the inclusion of the vessel for further use.

### 4.4. Serum-Induced Vasoconstriction

Serum-induced constriction experiments were conducted on omental arteries using gradually increased incubation with 2–20% of serum (trials demonstrated that the maximal constriction plateau was reached by a concentration of 20% serum). The serum was pooled from pregnancies complicated by preterm preeclampsia or term preeclampsia, as well as from corresponding gestation-matched normotensive controls who delivered at term. Following confirmation of artery integrity, each serum sample was added to the myograph bath in 2% concentration increments (120 µL), added approximately every 2 min (when the response to the previous dose plateaued), and the vascular response was measured with each addition. Changes in constriction were normalised to the pre-determined maximum constriction induced by KPSS.

### 4.5. Treatment with the Proton Pump Inhibitor Esomeprazole to Determine Effects on Vasorelaxation

Omental arteries collected from normotensive pregnancies were pre-constricted with serum to the calculated 50–70% of their total KPSS constriction; serum concentrations between 4–10% of preterm preeclamptic or control serum were required for this submaximal pre-constriction. Following this, cumulative half-log doses of 0.1 µM to 100 µM of esomeprazole sodium (Abcam, Cambridge, UK) or vehicle (water) were added every 2 min to measure vascular reaction via trace changes, which generated a dose–response curve. Values were expressed as a percentage of the maximum relaxation induced by bradykinin.

### 4.6. Statistical Analysis

Maternal sFLT-1, sENG, and ET-1 (ELISA) data were assessed for normal distribution and either an unpaired *t*-test or Mann-Whitney test was used to test the differences between the two groups (either preterm preeclamptic and preterm controls, or term preeclamptic and term controls).

Myograph constriction and relaxation curves were produced using non-linear regression (log[agonist] vs. response—four parameters or [agonist] vs. response—four parameters, as appropriate). Differences between responses to the agonist at each concentration were tested for significance using mixed-effects analysis with Šidák correction for multiple comparisons. The analysis of the area under the curve (AUC), maximum constriction, and (log)EC50 were tested for normal distribution and statistically tested as appropriate; the data were tested with an unpaired (parametric) *t*-test.

*p*-values < 0.05 were considered significantly different and all data are expressed as mean ± SEM. Statistical analysis was performed using GraphPad Prism 8.4.3 software (La Jolla, CA, USA).

## Figures and Tables

**Table 1 ijms-23-10852-t001:** Maternal clinical characteristics of serum samples from pregnancies complicated with preterm (early-onset) or term (late-onset) preeclampsia and gestation-matched normotensive control serum samples.

	Preterm Control	Preterm Preeclamptic	*p* Test	Term Control	Term Preeclamptic	*p* Test
** *n* **	10	10		7	4	
**Maternal age (years)** (SD)	28.70 (4.42)	31.10 (3.21)	0.182	33.29 (5.88)	33.75 (2.63)	0.886
**Body mass index; BMI (kg/m^2^)** [IQR]	24.00 [23.25, 25.75]	31.45 [27.25, 36.77]	0.004	32.60 [26.70, 34.00]	29.45 [24.87, 34.08]	0.849
**Gestation at delivery (weeks)** [IQR]	39.43 [39.00, 40.11]	28.64 [26.93, 30.79]	<0.001	39.00 [38.86, 39.07]	37.22 [37.11, 37.29]	0.008
**Gestation at blood collection (weeks)** (SD)	28.81 (2.49)	28.52 (2.57)	0.794	38.80 (0.38)	37.18 (0.14)	<0.001
**Highest systolic blood pressure during admission, including postpartum (mmHg)** (SD)	125.33 (8.47) ^	170.60 (14.03)	<0.001	117.14 (14.10)	152.50 (22.17)	0.010
**Highest diastolic blood pressure during admission, including postpartum (mmHg)** (SD)	79.11 (4.86) ^	97.20 (9.50)	<0.001	71.43 (10.29)	96.50 (12.61)	0.006
**Birth weight (g)** (SD)	3454.50 (470.51)	1049.60 (415.02)	<0.001	3728.57 (483.99)	2567.50 (422.64)	0.003
**Parity no (%)**						
0	5 (50.0)	9 (90.0)		1 (14.3)	2 (50.0)	
1	3 (30.0)	1 (10.0)		4 (57.1)	2 (50.0)	
2	2 (20.0)	0 (0.0)		2 (28.6)	0 (0.0)	
**Mode of delivery (%)**						
Vaginal	8 (80.0)	0 (0.0)	0.001	0 (0.0)	0 (0.0)	
Caesarean section	2 (20.0)	10 (100.0)		7 (100.0)	4 (100.0)	

Patient characteristics were analysed by comparing the preeclamptic groups to their gestation-matched controls. Statistical analysis for patient characteristics was performed using RStudio Software Version 3 (Build 458), PBC. ^ Highest SBP and DBP during admission was not recorded for one patient.

## Data Availability

Data available upon reasonable request from the corresponding author.

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
