# Peer review of "Serum Collected from Preeclamptic Pregnancies Drives Vasoconstriction of Human Omental Arteries—A Novel Ex Vivo Model of Preeclampsia for Therapeutic Development"

_ijms, 2022, doi:10.3390/ijms231810852_

Round 1

Reviewer 1 Report

This paper describes an interesting finding of Esomeprazole reducing pre-eclamptic serum dependent vasoconstriction in normotensive omental arteries in a novel ex-vivo model of pre-eclampsia. There are currently no treatments for pre-eclampsia other than delivery, and a great need for innovative approaches to identifying potential therapeutics such as repurposing of drugs. The authors have described their model in this well-written paper with some interesting initial results. I have minor comments on the manuscript as follows: 

Introduction: 

A more recent review on pre-eclampsia has recently been published in the NEJM (https://www.nejm.org/doi/full/10.1056/NEJMra2109523) which could be cited. The authors could be clearer regarding the proposed pathogenesis underlying early vs late onset pre-eclampsia and final common pathway leading to maternal vasoconstriction and endothelial dysfunction. 

Results: Section 2.2 and 2.3 - please clarify number of individuals/number of repeats per experiment in the text so that this is clear to the reader.

Conclusions:

3rd paragraph "Here, as expected we confirmed that serum obtained from pregnancies complicated by both preterm and term preeclampsia had significantly higher concentrations of the ET-1, sFlt-1 and sENG, compared to gestation matched controls." This is not supported by data - please update to reflect different findings for preterm and term pre-eclampsia.

4th paragraph: It is surprising that both control and pre-eclamptic serum induced vasoconstriction of the omental resistance arteries. Could the authors suggest any rationale as to why this might be? How do omental arteries compare to arteries used in other ex vivo models of pre-eclampsia? Further discussion and comparison to broader literature could be helpful. 

General comment - I would like to see discussion of the clinical trial data regarding esomeprazole use for treatment of pre-eclampsia in the discussion; do the reported data shed any light on what might be an optimal dosing strategy? 

Author Response

Reviewer 1

This paper describes an interesting finding of Esomeprazole reducing pre-eclamptic serum dependent vasoconstriction in normotensive omental arteries in a novel ex-vivo model of pre-eclampsia. There are currently no treatments for pre-eclampsia other than delivery, and a great need for innovative approaches to identifying potential therapeutics such as repurposing of drugs. The authors have described their model in this well-written paper with some interesting initial results. I have minor comments on the manuscript as follows: 

 1. Introduction: 

A more recent review on pre-eclampsia has recently been published in the NEJM (https://www.nejm.org/doi/full/10.1056/NEJMra2109523) which could be cited. The authors could be clearer regarding the proposed pathogenesis underlying early vs late onset pre-eclampsia and final common pathway leading to maternal vasoconstriction and endothelial dysfunction. 

RESPONSE: We thank the reviewer for generously taking their time to review our manuscript. We have included the citation as suggested (please see Page 1, Line 37) . So too, we have re-worded sections of the introduction to clarify pathogenesis underlying early compared to late onset preeclampsia (as below).

Please see Page 1-2, Lines 43-57 (changes are underlined):

“A further classification that infers the severity of the disease has been established; early-onset (preterm) preeclampsia is when delivery is necessitated prior to 34 weeks gestation [7]. There is an intersect in the culmination of preterm and term preeclampsia; both result in hypertension and often with end-organ injury and/or fetal growth restriction. However, there are differences in aetiology, risk factors and outcomes are apparent when they are stratified [8]. Compared to term preeclampsia, there is a higher probability of adverse birth outcomes when preeclampsia arises at a preterm gestation.

Inadequate placentation is thought to be associated with severe early onset preeclampsia [9]; poor remodelling of the spiral arteries leads to perturbations in blood flow to the placenta, resulting in placental dysfunction. The injured and stressed placenta releases antiangiogenic and proinflammatory mediators into the maternal circulation, where they neutralise proangiogenic factors, causing serious widespread endothelial dysfunction and systemic vasoconstrictor effects. Endothelial dysfunction and systemic vasoconstriction can compromise blood flow to major organ systems and the uteroplacental interface”

  1. Results: Section 2.2 and 2.3 - please clarify number of individuals/number of repeats per experiment in the text so that this is clear to the reader.

RESPONSE: We thank the reviewer for their suggestion. We have now included number of individuals in each of the experiments as below.

Section 2.2 - Please see Page 4, Lines 130-133:

Notably, there was no significant difference in the percentage vasoconstriction between the preeclamptic and control treatments at any concentration of serum, in both the preterm (Figure 2A; preeclamptic n=6, control n=5) and term (Figure 2B; preeclamptic n=4, control n=4) groups.”

Section 2.3 - Please see Page 6, Line 166-169:

“…there was no significant difference in the percent relaxation with increased concentrations of esomeprazole when compared to the vehicle (Figure 3A, n=3). However, analysis of the area under the curve demonstrated a significant difference between esomeprazole and vehicle treated vessels (Figure 3B; p=0.039; n=3).”

  1. Conclusions: 3rd paragraph "Here, as expected we confirmed that serum obtained from pregnancies complicated by both preterm and term preeclampsia had significantly higher concentrations of the ET-1, sFlt-1 and sENG, compared to gestation matched controls." This is not supported by data - please update to reflect different findings for preterm and term pre-eclampsia.

RESPONSE: We thank the reviewer for pointing this out. We have since clarified:

Please see Pages 7-8, Lines 236-241, as below (changes underlined):

“Here, as expected we confirmed that serum obtained from pregnancies complicated by preterm preeclampsia had significantly higher concentrations of ET-1, sFlt-1 and sENG, compared to gestation matched controls. Whereas only sFlt-1 was significantly increased in term preeclamptic samples, with trends of an increase observed for ET-1 and sENG. Increased sample size would of likely yielded a significant increase in term preeclampsia also [35, 36]."

  1. Levine, R. J.; Lam, C.; Qian, C.; Yu, K. F.; Maynard, S. E.; Sachs, B. P.; Sibai, B. M.; Epstein, F. H.; Romero, R.; Thadhani, R.; Karumanchi, S. A., Soluble Endoglin and Other Circulating Antiangiogenic Factors in Preeclampsia. New England Journal of Medicine 2006, 355, (10), 992-1005.
  2. Taylor, R. N.; Varma, M.; Teng, N. N. H.; Roberts, J. M., Women with Preeclampsia have Higher Plasma Endothelin Levels than Women with Normal Pregnancies. The Journal of Clinical Endocrinology & Metabolism 1990, 71, (6), 1675-1677.
  1. 4th paragraph:
    1. It is surprising that both control and pre-eclamptic serum induced vasoconstriction of the omental resistance arteries. Could the authors suggest any rationale as to why this might be?

RESPONSE: Both sera collected from preeclamptic, and normotensive control patients would contain factors that induce vasoconstriction (as demonstrated by our findings in Figure 1), given we expect that the circulation would contain a level of constricting factors to maintain normal homeostatic vascular tone. While somewhat surprising, the vasorelaxation for the experiments testing esomeprazole is therefore particularly interesting.

  1. How do omental arteries compare to arteries used in other ex vivo models of pre-eclampsia? Further discussion and comparison to broader literature could be helpful. 

RESPONSE: As suggested, we have now provided further discussion regarding other arteriole models in ex vivo models of preeclampsia; this includes human myometrial and chorionic arteries, in addition to small animal mesenteric vessels.

Please see Page 8, Lines 254-263:

“This study utilised human omental arteries, which are considered resistance arteries (which contribute to changes in blood pressure), and model systemic vascular changes. We believe that these are the best vessel type to address the systemic vasoconstriction and hypertension featured in preeclampsia. Further, omental arteries are relatively easily to access, being collected during caesarean section. However, assessing the effect of serum collected from pregnancies on other vessels including myometrial and chorionic arteries may also be of interest to identify local organ effects on the uterus and placenta. Assessing the effect of serum on mouse/rat systemic mesenteric arteries, may give insight into the differences between human and rodent vasculature – of relevance to animal models of preeclampsia.”

  1. General comment - I would like to see discussion of the clinical trial data regarding esomeprazole use for treatment of pre-eclampsia in the discussion; do the reported data shed any light on what might be an optimal dosing strategy? 

RESPONSE: We thank the reviewer for their comments. The clinical trial of 40mg esomeprazole daily did not prolong gestation of pregnancies complicated by preeclampsia (Cluver et al. Am. J. Obstet. Gynecol 2018) which was the primary outcome measured. From the clinical trial we concluded from the pharmacokinetic analysis that circulating levels of esomeprazole were likely too low, and perhaps higher concentrations may have exerted beneficial effects.

In our current study, we performed a dose-response curve with esomeprazole in our novel model. Our aim was to investigate whether esomeprazole overall could induce vasorelaxation, which was determined through the area under the curve and maximum relaxation parameters. However, we did not have the statistical power to specifically determine differences with each dose of esomeprazole used. Thus we present the overall data demonstrating a vasorelaxant effect in the arteries pre-constricted with the preeclamptic serum treated with esomeprazole.

Reviewer 2 Report

It is a very interesting study, which aims to create a new ex vivo model of preeclampsia, for therapeutic development.

Describes the development of an ex vivo model using serum from pregnant women with preeclampsia, in the endothelium of arteries from patients with a normal pregnancy to evaluate therapeutic interventions, which seems to be very promising, but as it is a new methodology, it should be described, with more detail.

It is worrying that the authors refer that one of the limitations of the study is the difficulty of its replication, which makes its use in the investigation of the different therapeutic candidates unfeasible.

Therefore, I consider that the manuscript requires greater precision in the methodology to consider that it meets the requirements of a new proposed methodology.

Notes for the authors

 Results

What was the rationale behind using 10 sera from patients with premature preeclampsia and 4 sera from patients with term preeclampsia?

 Discussion

The response to vasodilation caused by esomeprazole will be the same in arteries with previous endothelial and vascular dysfunction, then with intact arteries (normal pregnancy). 

Heparin was shown to interfere with the proposed methodology. 

It will be feasible to use the proposed preclinical model, using arteries from pregnant women complicated with premature and term preeclampsia. 

Preclinical models do not evaluate therapeutic efficacy, what they evaluate is an effect. 

They must specify the first limitation regarding the amount of serum necessary to carry out the proposed model. 

One of the characteristics of science is replication, and you argue that one of the limitations of the proposed model is the difficulty of its replication, this limitation indicates that it is unlikely that the proposed model can be used in the evaluation of different therapeutic candidates. 

Materials and methods

As this is a new methodology proposed to test the effect of different therapeutic candidates, it is necessary to specify the methodology in greater detail, so that it can be replicated by other researchers, for example, the amount of blood drawn is not specified, such as the amount of serum used in each of the proposed tests, etc. 

Why is the number of serum samples from patients with early and late-onset preeclampsia not equal to the number of sera reported in the abstract? 

The authors need to specify the amount of blood collected because they argue that one of the limitations of the proposed assay is the amount of serum that should be used. 

It should include the accuracy and precision of each of the tests, as well as the detection ranges.

What are the intra-assay and inter-assay variability?

What is the amount of serum used to perform each of the proposed determinations? 

How many human omental arteries are needed to assess vascular reactivity?

What are the volumes and times used for the evaluation of vascular reactivity? 

Based on what argument was a 2-20% serum concentration used to induce vasoconstriction and no other concentration or 100% serum? 

Again, what is the argument for using serum concentrations between 4-10% premature preeclampsia or control serum to constrict blood vessels and no other concentrations?

Author Response

  1. It is a very interesting study, which aims to create a new ex vivo model of preeclampsia, for therapeutic development.

Describes the development of an ex vivo model using serum from pregnant women with preeclampsia, in the endothelium of arteries from patients with a normal pregnancy to evaluate therapeutic interventions, which seems to be very promising, but as it is a new methodology, it should be described, with more detail.

It is worrying that the authors refer that one of the limitations of the study is the difficulty of its replication, which makes its use in the investigation of the different therapeutic candidates unfeasible.

Therefore, I consider that the manuscript requires greater precision in the methodology to consider that it meets the requirements of a new proposed methodology.

RESPONSE: We thank the reviewer for their time in reviewing our manuscript.  With the changes detailed in the responses below, we have provided further description of the precise methodology of the new model as suggested.

To clarify, we do not state a limitation of this study/model is a difficulty of replication. We suggest not all laboratories (nor many) would have access to such prized serum samples at an appropriate volume, and thus this model may be difficult to replicate in other laboratories. This could also be seen as a strength of our team, that we are able to model this.  However, given the confusion we have re-worded the section for clarity.

Please see Page 8, Lines 281-286 – which now reads:

“Whilst these novel preclinical models of preeclampsia provide physiologically relevant assays for the assessment of therapeutic efficacy, we recognise there may be difficulties for other researchers to establish this model in their own laboratories, especially given that collection of serum from early onset preeclamptic patients is only attainable within specific clinical settings. Additionally, a significant amount of serum is required to have sufficient power in the study.”

Notes for the authors

 Results

  1. What was the rationale behind using 10 sera from patients with premature preeclampsia and 4 sera from patients with term preeclampsia?

RESPONSE: The rationale behind different number of samples was the availability of patient serum samples at the time that these experiments were conducted. We have greater access to severe early onset serum, and hence there was more of this available.

Discussion

  1. The response to vasodilation caused by esomeprazole will be the same in arteries with previous endothelial and vascular dysfunction, then with intact arteries (normal pregnancy).

 RESPONSE: We thank the reviewer for highlighting this. We have used normal, term omental arteries in this study – but we acknowledge that arteries collected from individuals with preeclampsia may respond differently.

We have considered this in our discussion, please see Page 9, Lines 295-298:

“An additional consideration is that the arteries used were obtained from term normotensive term pregnancies.  Thus, it is important to consider whether these vessels have the same vasoactive potential compared to vessels from pregnancies complicated by preeclampsia or hypertension.” 

  1. Heparin was shown to interfere with the proposed methodology.

RESPONSE: We have mentioned in the manuscript that heparin is used in models that use plasma to prevent coagulation (Page 7, Lines 222-225). Heparin was not used in these experiments as we used serum.

  1. It will be feasible to use the proposed preclinical model, using arteries from pregnant women complicated with premature and term preeclampsia.

RESPONSE: We thank the reviewer for this comment. We agree that this would be of interest. However, as we are using systemic maternal omental arteries, we would not expect the vessel response to change as much across gestation compared to the myometrial or chorionic arteries which are directly/locally affected by the pregnancy. We would be more interested in investigating vessels collected from pregnancies complicated by preeclampsia as stated in response  3. 

  1. Preclinical models do not evaluate therapeutic efficacy, what they evaluate is an effect.

RESPONSE: We thank the reviewer for their suggestion. We have made the changes as below.

Please see Page 8, Lines 281-282 (changes are underlined):

Whilst these novel preclinical models of preeclampsia provide physiologically relevant assays for the assessment of therapeutic effects…”

  1. They must specify the first limitation regarding the amount of serum necessary to carry out the proposed model.

RESPONSE: With the following, where we have detailed the amount of serum necessary, we trust further clarity in the precision of the methodology is provided.

Please see Page 11, Lines 395-396: “Serum-induced constriction experiments were conducted on omental arteries using a gradual increased incubation with 2-20% of serum…”, and Lines 400-403: “each serum sample was added to the myograph bath in 2% incremental concentrations (120µL), added approximately every 2 minutes (when response to previous dose plateaued), the vascular response was measured with each addition.”

  1. One of the characteristics of science is replication, and you argue that one of the limitations of the proposed model is the difficulty of its replication, this limitation indicates that it is unlikely that the proposed model can be used in the evaluation of different therapeutic candidates.

RESPONSE: We understand the reviewers concern and have further clarified the limitation as above (Response 1), where the limitation was more due to the availability of samples, rather than technical replication – we do not think there is an issue with reproducibility. With these, we believe the proposed model remains novel and strong.

  1. You could use pregnant serum – clarify that it may be difficult to replicate where preeclamptic serum is unavailable.

RESPONSE: We agree with the reviewer and as detailed in Response 1, availability of preeclamptic serum may be a issue for some. Hence, pregnant serum may be an alternative, however we note that in our study, esomeprazole only dilated vessels that were pre-constricted with the preeclamptic serum. We trust that the above corrections listed in Response 1 have clarified this. 

  1. As this is a new methodology proposed to test the effect of different therapeutic candidates, it is necessary to specify the methodology in greater detail, so that it can be replicated by other researchers, for example, the amount of blood drawn is not specified, such as the amount of serum used in each of the proposed tests, etc.

Why is the number of serum samples from patients with early and late-onset preeclampsia not equal to the number of sera reported in the abstract?

RESPONSE: We thank the reviewer for their comments and have added more detail into our methodology as described in the following responses.

We thank the reviewer for pointing out the serum sample numbers. We have now corrected the n number of serum samples from patients with early and late-onset preeclampsia stated in the abstract. Subsets of these samples were used in the myograph assessments presented in Figures 2 and 3.

See Page 1, Lines 21-23, which states (changes underlined):

Serum was collected from pregnancies complicated by preterm preeclampsia (birth <34 weeks’ gestation, n=16); term preeclampsia (birth >37 weeks’ gestation, n=5) and healthy gestation matched controls (preterm n=16, term n=12).”

  1. The authors need to specify the amount of blood collected because they argue that one of the limitations of the proposed assay is the amount of serum that should be used.

RESPONSE: We thank the reviewer for noting this. The methods (Section 4.1.1. Serum collection) now detail the specific amount of blood collected from each patient.

See Page 9, Lines 337-339, which states (changes underlined):

Blood (8ml) was collected into serum tubes (Serum Separator Tube (SST®); gold cap), were gently inverted and subsequently centrifuged at 1500g for 10 min.”

  1. It should include the accuracy and precision of each of the tests, as well as the detection ranges. What are the intra-assay and inter-assay variability? What is the amount of serum used to perform each of the proposed determinations?

RESPONSE: We thank the reviewer for their question. All ELISA kits used in this study are commercially available and pre-validated, which is why they were chosen for these research studies. All ELISAs were performed according to manufacturers instructions - we have added additional information on the kits including the commercial product name and catalogue number. We have included the dilutions used, and have clarified the intra-assay coefficients for the purchase ELISA kits is under 10% as below. All duplicates showed consistency and precision in pipetting. All samples were run on one plate with one standard curve for each ELISA, hence there is no inter-assay variability.

Please see Page 11, Lines 365-370 which states (changes underlined):

Serum levels were measured using the Human VEGFR1/Flt-1 DuoSet ELISA (1/20 dilution; DY321B, R&D systems, Minneapolis, MN, USA), Human Endoglin/CD105 DuoSet ELISA (1/15 dilution; DY1097, R&D systems) and the Endothelin-1 Quantikine ELISA Kit (neat serum; DET100, R&D systems, sensitivity: 0.207 pg/mL), according to manufacturer’s instructions (intra assay coefficients of variations under 10%).”

  1. How many human omental arteries are needed to assess vascular reactivity? What are the volumes and times used for the evaluation of vascular reactivity?

RESPONSE: The sample number depends on the experimental design, and samples used. In our experiments, at least n=3 was sufficient to achieve statistical significance. Further, we used a paired experimental design between control and treatment which strengthens our results. As we detail in the methods (Page 11, Lines 399-403), following each dose of serum added (2%, or 120uL), the constriction response was measured after 2 minutes – this timing was chosen to allow the vessels to reach a plateau of response before the next dose was added.

  1. Based on what argument was a 2-20% serum concentration used to induce vasoconstriction and no other concentration or 100% serum?

RESPONSE: We thank the reviewer for their query. The constriction produced by the serum reached a maximum plateau at 20%; this was optimised in previous dose response experiments where higher concentrations of serum was added into the chamber. The following has been added to clarify this, please see Page 11, Lines 395-397:

“Serum-induced constriction experiments were conducted on omental arteries using a gradual increased incubation with 2-20% of serum (trials demonstrated that the maximal constriction plateau was reached by a concentration of 20% serum).”

  1. Again, what is the argument for using serum concentrations between 4-10% premature preeclampsia or control serum to constrict blood vessels and no other concentrations?

RESPONSE: It is common practice in wire myography to pre-constrict arteries to 50-70% of their maximum constriction (induced by a high potassium salt solution (KPSS)), prior to testing vasorelaxation. A range of 4-10% was required to reach this optimal constriction in all the vessels used. This has been stated in the methods as follows, please see Page 12, Lines 409-412:

“Omental arteries collected from normotensive pregnancies were pre-constricted with serum to the calculated 50-70% of their total KPSS constriction; serum concentrations between 4-10% of preterm preeclamptic or control serum were required for this submaximal pre-constriction.”